# The Role of Technology in Physical Education Teaching in the Wake of the Pandemic

Diana Marín-Suelves [1], Jesús Ramón-Llin [2,*] and Vicente Gabarda [1]

1 Department of Teaching and School Organization, University of Valencia, 46010 Valencia, Spain
2 Department of Teaching of Physical Education, Artistic and Music, University of Valencia, 46010 Valencia, Spain
* Correspondence: jesus.ramon@uv.es

**Abstract:** Physical education is seen as an essential subject for the development of healthy habits and well-being, in line with Sustainable Development Goal 3. Furthermore, the impact of technology on all aspects of life is now an undeniable reality. The field of education is no exception, and digitalisation has undoubtedly been accelerated by the emergency situation resulting from the COVID-19 pandemic. This paper aims to analyse the scientific production related to the field of physical education, technology, and the pandemic from a double perspective. From a search in Scopus, 86 articles were selected for analysis. A bibliometric approach was used to identify the variables of impact, collaboration, production, and dissemination. While the content analysis allowed us to delve deeper into the topics most frequently chosen by researchers, we found that the articles focused both on the circumstances experienced by practising teachers and on the adaptations made in the teaching/learning process with trainee teachers and students at different stages of education. Thus, technology has emerged as a fundamental tool in physical education during the pandemic, making it possible to develop or maintain better health and learning.

**Keywords:** technology; pandemic; physical education

## 1. Introduction

In recent years, there have been several issues that have shaped the education policies of different education systems.

On the one hand, the Sustainable Development Goals (SDGs) propose a framework of recommendations that can help governments set their policies in different areas to respond to current global needs. Based on a comprehensive approach, this proposal includes issues such as climate change and limited natural resources, reducing inequalities and achieving social inclusion, or promoting and ensuring healthy habits for a better life. Thus, physical activity is seen as a resource for achieving health and well-being, and in the field of education, physical education will play a fundamental role in achieving these healthy habits.

On the other hand, the pandemic has caused an abrupt disruption in our lives, changing the way we develop academically, professionally, and socially. In the field of education, after the shutdown of educational institutions, professionals had to adapt their practice to the new reality and implement strategies to continue teaching and learning through technology. Thus, regardless of educational level or subject, tools for designing online learning scenarios began to be implemented, materialising in the use of different devices, platforms, and software.

An unresolved question, however, is how has the subject of physical education been affected by this phenomenon? In this article, we try to answer this question by analysing the specific role that technology has played in the teaching of physical education during and after the pandemic.

### 1.1. The COVID-19 Pandemic and Its Impact on Formal Education

As mentioned above, the situation resulting from the COVID-19 pandemic required an urgent abandonment of physical presence in the classroom [1] and a radical and immediate adaptation of training scenarios [2]. It was therefore necessary to rethink issues such as timetables, resources, and methodologies, as well as the roles and responsibilities arising from this scenario, for which the various educational actors seemed unprepared in terms of methodological and instrumental training and digital literacy.

The general response has been technological [3], taking advantage of the potential that digital devices and resources can offer for the generation of delocalised training actions [4,5] and the creation of hybrid or online training spaces that allow learning processes to continue [6]. This digital transformation has not been free of some critical issues, such as the digital divide, which has led, for example, to the neglect of the most vulnerable sections of the population [7,8], the lack of competences of teachers to design and implement their educational work in this new scenario [9], or the monolateral strategies generated by the management of educational institutions and public administrations [10]. In many cases, this process of adaptation simply took the form of transferring what had been designed for the classroom to a digital context, without questioning its suitability or reflecting on the pedagogical foundations of the new format [11].

### 1.2. The Impact of the Pandemic on the Learning and Teaching of Physical Education

This scenario, characterised by a technological component in the training process that has become both compulsory and urgent, has led to the emergence of different pedagogical responses on the part of teachers in different disciplines, depending on issues such as the subject area, the educational stage, the personal characteristics of each teacher, their methodological and digital skills, or the policies prescribed by public administrations or educational institutions.

However, physical education is a particularly relevant subject to study, both in terms of its curricular content and the methods and resources commonly used to implement it [12]. This discipline is usually associated with practical content, sometimes involving the use of specific physical materials, the use of which requires face-to-face supervision. In addition, confinement policies have often led to an increase in sedentary lifestyles and a deterioration in the physical condition of students [13], an aspect that is clearly detrimental to the discipline itself.

We can therefore see that there has been some concern, at both academic and social levels, to promote habits that maintain physical fitness, strengthen the respiratory and immune systems, and have a positive effect on mental health [14], as well as the development of protocols to adapt the teaching of the discipline to the so-called "new normal", working on health promotion and care during class time, and creating healthy habits and a more consistent adherence to exercise [15].

The point is that, despite these handicaps, physical education has had to adapt its principles and practical application to a context of confinement, incorporating methodologies such as gamification or the flipped classroom [16], which harness the potential of technology to develop training activities with a more dynamic and participatory approach [17]. In other disciplines, these tools have shown very positive results in adapting the training process [18] to different modalities, promoting student motivation [19] due to the interactivity of the teaching and learning processes involved, and improving attention [20] (due to the visual component of the resources), performance [21], and the development of different skills [22] for the range of possibilities offered by technology.

Thus, studies such as [23] or [24] highlight the benefits that these methods can bring to learning, concluding that they can improve student performance and competence development, and provide critical, meaningful, ubiquitous, transformative, and motivating experiences. This provides further evidence of their positive impact on educational processes, in line with previous studies such as [25], which supported mobile technology; [26], which analysed video tutorials as a teaching tool; and [27], which showed the impact of

video games in physical education. These studies also confirmed that technology promotes the development of physical and psychological skills, enhances creativity and personal involvement, and facilitates an individualised pace of learning.

Based on this approach, the aim of this article is to analyse, from a bibliometric and content analysis point of view, the impact of technology on learning in physical education during and after the COVID-19 pandemic.

## 2. Materials and Methods

This work was carried out by combining a bibliometric study, which quantifies scientific production, with a content analysis, which allows an in-depth approach to the object of study.

The Scopus database was used to identify the documents to be analysed. The following search terms were used in combination with Boolean operators: technology OR digital AND pandemic OR COVID AND "physical education".

No filters were used except for a restriction on the type of document, including only articles. All scientific documents of this type that focused on the use and impact of technology in physical education during the COVID-19 pandemic were selected for analysis. Exclusion criteria included review or instrument validation articles and topics unrelated to the subject of the study.

The PRISMA method [28] was used to systematise the process. The first phase consisted of identifying documents related to the topic by searching for terms in the title, abstract, or keywords. The second phase consisted of screening by applying the established filters. The third phase of eligibility was based on the evaluation of full text articles. Finally, we decided to include a total of 86 articles on the role of technology in physical education during the pandemic. Figure 1 illustrates the process.

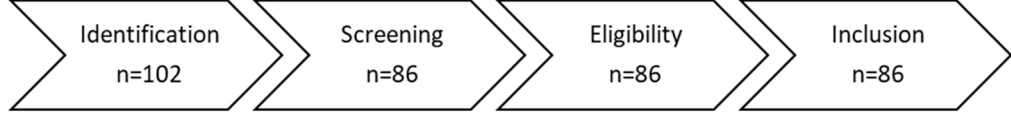

**Figure 1.** Flowchart.

The variables taken into account are shown in Table 1.

**Table 1.** Variables.

| Analysis | Variable | Category |
|---|---|---|
| Bibliometric | Productivity | Date of publication<br>Country<br>Areas<br>Language |
| | Collaboration | Authorship<br>Networks |
| | Impact | Number of citations<br>Typology of producers<br>Sources |
| | Dispersion | Zones |
| Content | Participants | Students<br>Future teachers<br>Teachers |
| | Technology | Teaching Modality |
| | Assessment | Benefits<br>Difficulties<br>Controversies |

Four indicators and three bibliometric laws (Price, Lotka, and Ford) were taken into account in the bibliometric analysis. Price's law states that the number of publications multiplies approximately every 10 or 15 years. Lotka's law suggests that the number of authors dedicated to a field of study allows the identification of a small number of specialised authors who are considered to be major producers. Finally, Bradford's law refers to the central position of a reduced number of journals in which publications on a subject accumulate.

The VosViewer software 1.6.19 version was used to plot the data.

## 3. Results

### 3.1. Bibliometric Approach

Productivity was analysed based on date of publication, location, field of expertise, and language used for knowledge transfer.

The COVID-19 pandemic crippled the world in early 2020. The impact of this virus on the lives of millions of people was enormous. Researchers in all fields continued to make great efforts, resulting in the publication of a large number of papers related to this virus. As shown in the figure below (see Figure 2), the increase in publications on technology, coronaviruses, and physical education was significant. The data referring to the current year (2023) are not represented in Figure 2, since the definitive data on publications in this year are not known.

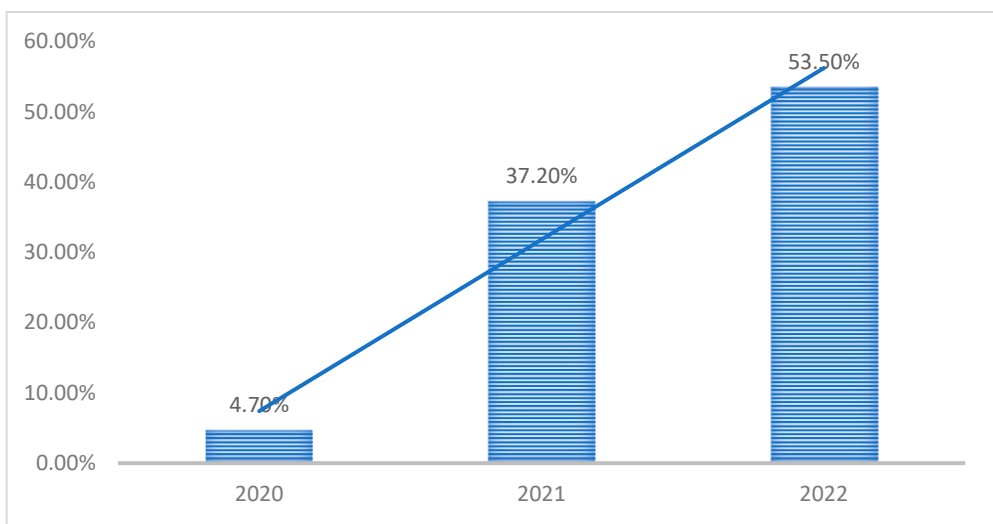

**Figure 2.** Evolution of the number of publications per year (2020–2022).

With regard to the countries of origin of publications in this field, Figure 3 shows the productivity of all countries with five or more articles. In addition, 40 other countries from the five continents have at least one publication on technology, physical education, and the coronavirus. In the Americas, a large number of producing countries stand out, such as the USA, Brazil, Chile, Argentina, and Colombia. In Europe, many countries also produced papers, with Spain being a major contributor, although other countries such as Italy, Belgium, Bulgaria, France, Finland, and Germany were also represented in the documents analysed. In Asia, contributions came mainly from China. In Oceania, Australia was the largest contributor. Meanwhile, the African continent is represented only by Tunisia, which is surprising given the number of countries on the continent.

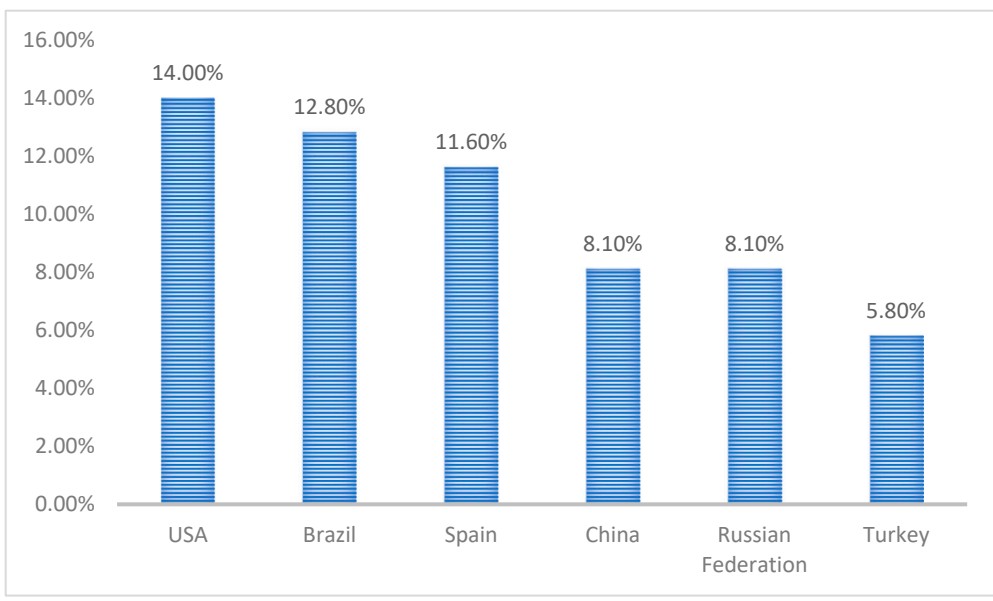

**Figure 3.** Countries with the highest productivity.

In terms of fields of knowledge, the presence of a significant number of publications in health sciences (37.8%), social sciences (33.2%), computer sciences (5.2%), and a wide range of other fields is particularly striking (see Figure 4), including fields such as psychology, engineering, neurosciences, and environmental sciences.

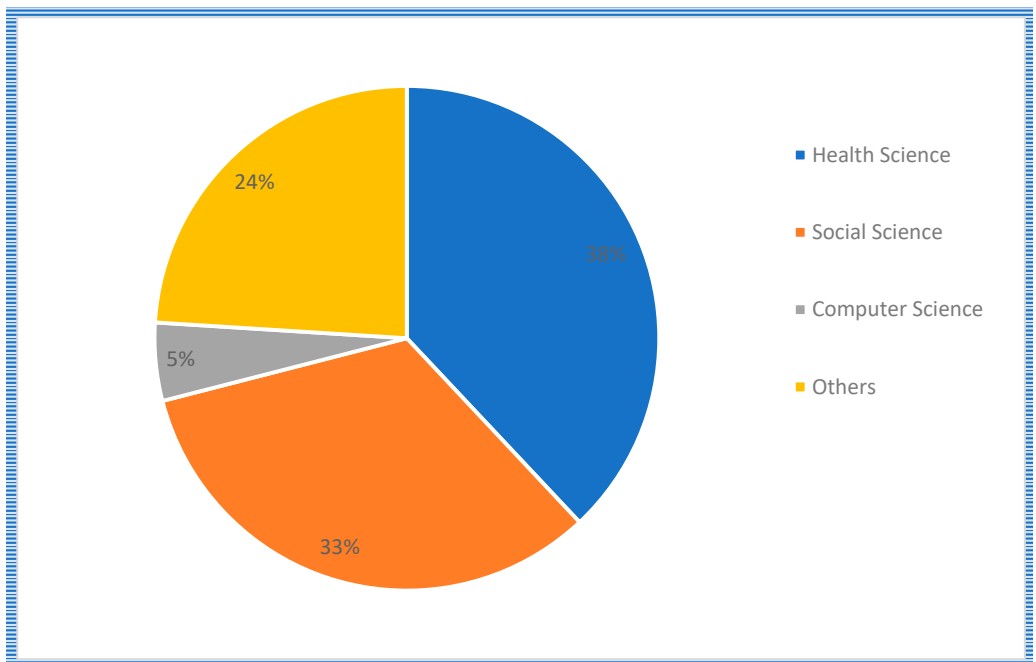

**Figure 4.** Areas of knowledge.

Finally, the language in which most of the articles analysed were written is English (75%), followed by Spanish (11.9%) and Russian (7.6%), although there were also articles published in other languages (see Figure 5).

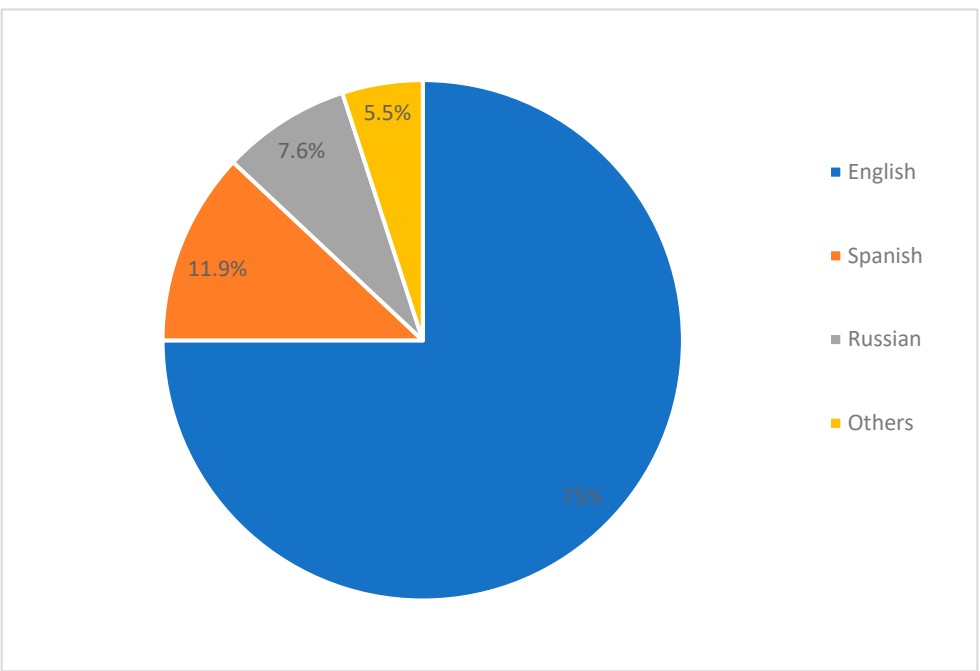

**Figure 5.** Language.

Secondly, collaboration was analysed by looking at authorship and the collaborative networks created. The degree of collaboration was very high, with the majority of articles (93.1%) written by more than one author.

In terms of collaborative networks (see Figure 6), there were papers written by researchers from several countries, such as [29], written by Brazilian, American, and Irish researchers, who analysed the impact of experience on collaboration and self-organisation from the perspective of physical education teachers. The most common, however, were papers by authors from the same institution, or by authors from different institutions within the same country.

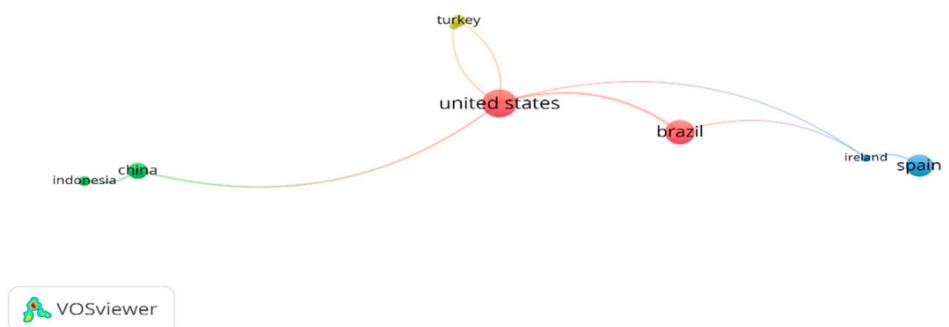

**Figure 6.** Collaborative networks.

The third variable analysed in the bibliometric study was impact, measured through the type of producer, the number of citations, and the sources.

In terms of the number of publications by the same author on the topic, there were no major producers; there were only two authors with three publications, namely Erin Centeio [30–32] and Allyson Carvalho de Araújo [33–35].

In terms of cumulative citations, only nine papers had ten or more citations. This can be explained by the time frame analysed, which was limited to three years, with the emergence of the COVID-19 pandemic as the starting point for the analysis. The following table shows the data for articles with more than 15 citations (see Table 2).

**Table 2.** Most-cited articles.

| Identification Data | Journal | SJR Education | Citations |
|---|---|---|---|
| [36] | *Physical Education and Sport Pedagogy* | Q1 | 41 |
| [32] | *Journal of Teaching in Physical Education* | Q1 | 29 |
| [37] | *Physical Education and Sport Pedagogy* | Q1 | 19 |
| [38] | *European Educational Research Journal* | Q2 | 18 |

Among the journals with the highest number of articles on technology, physical education, and the coronavirus, the following stand out with five or more articles (see Table 3):

**Table 3.** Impact of journals.

| Journal | SJR | Articles | Citations | Impact |
|---|---|---|---|---|
| *Sustainability* | Q2 | 9 | 32 | 3.55 |
| *Movimento* | Q3 | 7 | 21 | 3 |
| *Retos* | Q3 | 5 | 12 | 2.4 |

Finally, the analysis of the dispersion through the zones created by taking into account the number of articles accumulated on a topic in the same journal allows us to confirm Bradford's law, with the core formed by only five journals and one-third of the articles analysed, while there are no differences between the other two zones (see Figure 7).

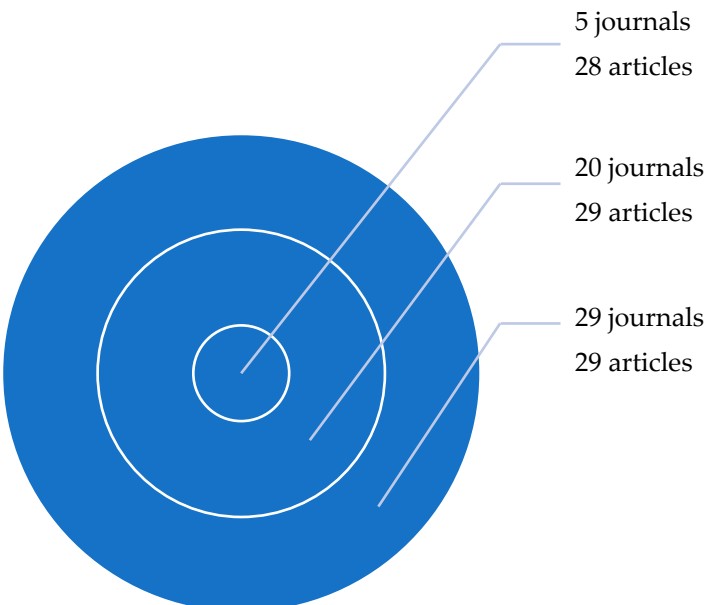

5 journals
28 articles

20 journals
29 articles

29 journals
29 articles

**Figure 7.** Dispersion zones.

*3.2. Content Analysis*

The following figure shows the most frequent terms in the analysed articles, with the size of the circles and words changing according to the number of times they were used, and their frequency of co-occurrence represented by the lines. The colours show the emergence of four clusters, with the blue cluster representing the axis of this work. This cluster includes terms such as physical education, technology, and COVID-19.

The green cluster is made up of words such as physical activity, students, distance learning, online teaching, and blended learning. Thus, there is a focus on the role of the

learner and on teaching/learning methods, which have gained importance in COVID-19 due to the exceptional circumstances of the global health crisis.

As for the red cluster, it seems to focus on the adult stage—regardless of the gender of the participants—and on physical education and training.

Finally, the yellow cluster is made up of concepts such as teaching, learning, and education, with the aim of avoiding a lost school year, but also considering the benefits that the physical component has for the integral development of people of different genders, ages, and locations (see Figure 8).

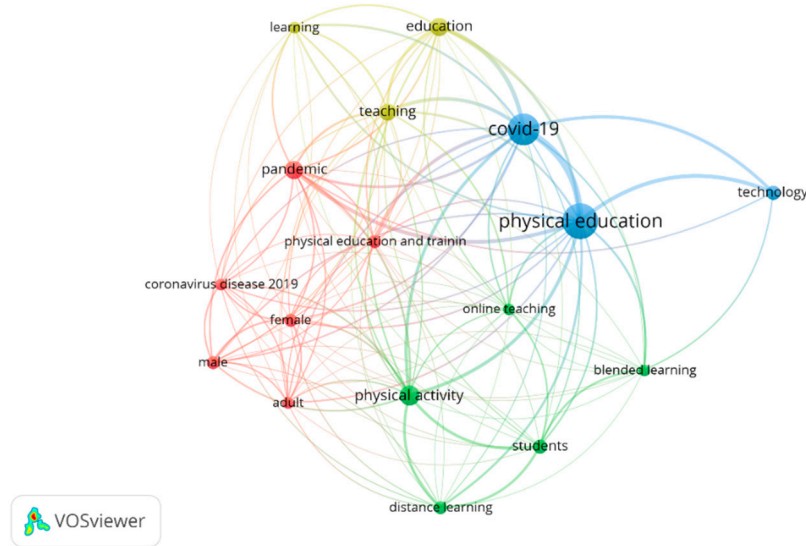

**Figure 8.** Clusters.

We now move on to the content analysis, first describing the type of participants, then the digital technologies used, and finally an analysis and evaluation of the results obtained.

### 3.2.1. Type of Participants

Regarding the type of participants studied, three main types can be identified. On the one hand, we found studies that focused on students [39,40]. For example, Antekolović and Kovačić [39] analysed university students from the University of Zagreb (Croatia), while [40] focused on the study of primary school students with special educational needs in the Dominican Republic.

Other studies focused on students who will be future physical education teachers [41,42]. For example, Almonacid-Fierro et al. [41] focused on conducting interviews with future teachers who were undertaking grade internships during the COVID-19 pandemic, while Rodríguez [42] analysed the case of a future primary school PE teacher who was in year 4 of her primary school education.

Finally, there are also a large number of studies that focused on active teachers [29,30,43–45]. For example, Asún-Dieste et al. [43] focused on Spanish university teachers who supervised Final Degree Projects in Physical Education, while other studies focused on teachers of Basic Public Education in Brazil [44], on teachers from two Brazilian Federated Institutes of Physical Education [29], on elementary and high school teachers who implemented online physical education in Indonesia [45], and on multilevel elementary, middle, and high school teachers in the United States [30].

### 3.2.2. Digital Technologies

The digital technologies used were diverse and fundamentally adapted to the type of teaching methodology or modality used. For example, Campos-Mesa et al. [46] used active methodologies based on flipped classroom models, using traditional explanatory videos or

augmented reality models with videos designed by the teacher and later uploaded to the Kaltura media platform for students to access.

Based on the physical literacy model and in order to improve motivation towards physical education, Blain et al. [47] used a gamified blended approach after the pandemic. On the one hand, they used videos that provided demonstrations and information for students to overcome motor challenges, and on the other hand, they also used independent narrative resources to contextualise the situations or set challenges in an escape-room type situation where students had to "beat a super villain", for example.

As mentioned above, studies were found where the teaching modality was online, as were others that used mixed modalities. For example, Mariano et al. [48,49] analysed an online teaching modality during the pandemic in which e-sports (sports video game competitions) such as Mobile Legends or Team Sports were used and the losers were penalised with additional physical exercise. Also in the online teaching modality, Kulkarni et al. [50] combined the use of different technologies in a model of enhanced online engineering education. They combined the use of the Go To webinar, the Udemy online learning platform, and the LearniCo mobile application for students. The combination of these tools allowed them to ensure that the activities were carried out by the students and to obtain statistical reports on learning and attendance. On the other hand, Shaowei et al. [51] used explanatory videos and related tasks in the online teaching modality for blended learning in physical education, while the offline teaching modality was dominated by interactions and discussion groups in which students contrasted their opinions or teachers created concept maps to structure knowledge.

### 3.2.3. Assessment of the Result of the Different Studies

When analysing the outcomes of interventions using different technologies, studies reported benefits in different areas. According to [52], one of the areas that improved during the pandemic was students' digital literacy in information and communication technologies, learning how to use the Internet, email, and other online communication tools.

In terms of social and emotional aspects, it was observed during the COVID-19 pandemic that students with greater knowledge of ICTs were more satisfied and motivated with online teaching [52], but a lack of social and emotional support for students was also reported, in addition to a lack of teacher training in the use of technology in online teaching [53]. In sport, the difficulties of teaching in a COVID-19 context led to an increase in burnout among coaches using a 100% online modality, which was lower among those using a blended modality [54].

Teaching difficulties affected teachers differently depending on their level of education. For example, ref [45], who analysed the teaching skills of physical education teachers, found that high school teachers had difficulties in initiating and organising learning and in implementing strategies to improve students' practice. On the other hand, primary school teachers had problems in finalising the syllabus and dividing the teaching tasks according to the students' abilities. In this line, other studies showed the problems of teachers' assessment of motor competence, but reported that an intervention and training in ICTs improved teachers' perception of competence [55].

Finally, regarding the health and motor skills of physical education students during the COVID-19 pandemic, the results are mixed. On the one hand, Paramitha et al. [56] pointed to the health benefits of online gymnastics for adults, but on the other hand, Rutkauskaite et al. [57] reported that students' physical activity decreased and their fitness deteriorated during the pandemic period. In particular, it was found that students with lower computer skills, those who spent more leisure time using ICTs, and older students were found to be less physically active and to have experienced the greatest decline in motor skills. A study by [58] also showed conflicting results, indicating greater obesity, especially in boys, but an improvement in vital capacity—which is related to cardiorespiratory function—in girls. The same study [58] showed improvements in both males and females in flexibility (sit

and reach test), muscular strength (standing long jump test), and also muscular endurance (pull-up test for males and sit-up test for females).

## 4. Discussion and Conclusions

There is no doubt that the pandemic has had an impact on the way we teach and learn. More than ever, technology has become a tool that allows us to address issues such as the lack of face-to-face interaction or the importance of being able to rethink training processes from a methodological point of view [17]. However, it should be stressed that in order to implement technology in an inclusive way, it is necessary to work on those aspects that the pandemic has brought to light, such as the digital divide, the lack of equipment or connectivity, and the lack of training of agents in digital matters [7,9].

With regard to the bibliometric laws, firstly, it is not possible to assess the compliance with Price's law (on the growth of scientific information) with the available data, since the period analysed in this study was limited to three years, because the subject was linked to the COVID-19 pandemic. Secondly, based on Lotka's law, which focuses on the relationship between the number of documents and the number of authors writing on a topic, no major producers were identified in this study, as in previous investigations [59]. Thirdly, the distribution of publications in this study is observed in concentric zones of decreasing productivity, following Bradford's law and in line with previous work, such as that of [27]. Other results obtained in this study are in line with previous work using this approach, such as the preponderance of publications belonging to the USA [60], the weight of the journal *Sustainability* [61], the predominance of works corresponding to the field of health sciences [62,63], or the high degree of collaboration between authors [64].

In terms of content analysis, the reviewed studies considered students [39,40] as participants in physical education and sport, as well as teachers in training [41,42] and active teachers [43,44]. Regarding the technologies used in both online [48,49] and blended [47,51] teaching, different platforms and webinars [46] stood out, as well as the use of augmented reality [46] or explanatory videos in flipped classroom methodologies [46] and gamification [47], mobile applications, or e-sports competitions [48]. When analysing the results, different effects were observed depending on the level of ICT competence of students and teachers [45,52], and the negative emotional impact of the situation was confirmed for both students [53,54] and teachers. Findings on health effects were mixed. For example, a side effect of confinement was that it affected people's health both physically [56,58] and emotionally [52,53]. Consequences such as a sedentary lifestyle [57], weight gain [58], changes in eating and sleeping habits [65], and diseases associated with an unhealthy lifestyle have also been reported [65]. On the other hand, the efforts of the medical and sports sectors to use telemedicine and social networks to promote physical activity and the gradual return to exercise and professional sport have also been analysed [65].

Finally, in order to ensure healthier lives and promote well-being at all ages (SDG 3), promoting physical activity habits in childhood has been linked to good physical activity habits in adulthood [66]. Physical activity is associated with a reduced likelihood of developing obesity and cardiovascular disease [67]. It is therefore advisable to create environments or spaces that are conducive to physical activity and have a good walkability index [68]. Following this line, physical education is presented as a key subject to create healthy physical activity habits [69], and for this reason many studies focus on improving students' motivation towards this subject [70].

However, the pandemic has had a major impact on distance learning methodologies, and this is true also in the case of physical education [71]. The digital literacy of teachers, trainers, students, and trainees has been key to the development of these processes during and after the COVID-19 crisis. For example, it is now commonplace to see a variety of programmes on television offering activities such as yoga, Pilates, or healthy exercise alternatives for all ages and fitness levels. In the case of physical education, there has also been an increase in the use of exer-games (video games in which physical exercises are performed to improve students' motivation and motor skills).

Finally, the work carried out allows us to analyse the usefulness and impact of the introduction of technology in physical education during and after the pandemic; however, there is a limitation, namely the use of only one database, in this case Scopus. In the future, we are considering the possibility of replicating this work using other databases and going a step further to carry out analyses linked to specific territories, since the digital divide exists and became even more evident during the pandemic. In addition, altmetrics can be introduced in future analyses for a more social approach to scientific production in this field.

**Author Contributions:** Conceptualization, V.G. and D.M.-S.; software and validation, D.M.-S. and J.R.-L.; formal analysis, D.M.-S., V.G. and J.R.-L.; data curation, D.M.-S. and J.R.-L.; results and conclusion, D.M.-S., V.G. and J.R.-L.; writing—original draft preparation, D.M.-S., V.G. and J.R.-L.; writing—review and editing, D.M.-S., V.G. and J.R.-L. All authors have read and agreed to the published version of the manuscript.

**Funding:** This research received no external funding.

**Data Availability Statement:** Data is available in https://crie.blogs.uv.es/2023/05/23/el-impacto-de-la-tecnologia-en-la-educacion-fisica/.

**Conflicts of Interest:** The authors declare no conflict of interest.

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
