# Peer review of "The Role of Technology in Physical Education Teaching in the Wake of the Pandemic"

_sustainability, doi:10.3390/su15118503_

Round 1

Reviewer 1 Report

1. What role has technology played in teaching physical education during the pandemic?

 2. The article identifies, at the level of the 86 articles selected from BDI Scopus and analysed, the technology applied in physical education classes in countries on all continents during the pandemic. It provides teachers with tools to use even after the pandemic in adapted contexts.

 3. The authors provide readers/specialists with a picture of the concerns related to the organization/conduct of physical education classes in the period 2020-2022 in different countries of the world, by analysing 86 articles.

4. The authors used methods appropriate to the intended purpose.

5. We believe that the authors have formulated their conclusions in line with the evidence and arguments presented and that they address the main question posed.

 6. The bibliographic references are appropriate: related to COVID 19, to the technologies used during the pandemic in physical education classes, etc.

Author Response

We welcome reviewer's comments and suggestions as they help us improve the quality of our work.

The text has been revised by a professional translator. The certificate issued as proof of the translation work and revision of the linguistic correction is attached.

The terminology and use of acronyms have been matched throughout the document, as requested.

References and citations have been reviewed, both in text and in the specific section.

Inclusion criteria have been introduced to facilitate understanding of the process followed. In addition, bibliometric laws have been defined.

The format of Tables and Figures has been modified, as well as the mention in text and the legends in which it was necessary.

In Figure 7, the adjustment to the data and its correction have been verified.

Improvements have been made to the text and changes to some phrases or terms that seemed to cause confusion.

Due to the very nature of the bibliometric approach, we have addressed all the dimensions that have emerged from the analysis of the selected documents, therefore, in this work we work from a global approach, but we take note so that in the future we can carry out specific work on those dimensions presented. .

The statements in the first lines are based on news in the press and the experience lived in the months in which the COVID-19 pandemic hit the hardest, therefore academic citations are not included.

The nuance of policies has been introduced in the right place, as we agree with this view.

Results of some countries such as New Zealand or Australia are not detailed because they are not countries that have greater production or impact than the work carried out in others.

Reviewer 2 Report

The topic is good but you should be more rigorous in your writing.

Remember that others will read your content and you should keep the same way of explaining certain ideas. For example, complete phrase - "pandemic" by "COVID-19 pandemic" - since in time there may be others and in other places. Try to unify the term, elsewhere in the document indicates: Covid-19 (abstract), covid-19 (page 8), COVID pandemic (page 8), COVID (page 11) or pandemic i.e. retain upper case for this type of virus.

You should also standardize some acronyms, for example PE should be Physical Education (page 3) or PA (page 11) ¿?, as well as the use of Sustainable Development Goals (SDG 3) or (Sustainable Development Goal 3) and others that are not identified. 

Do not mix citation formats, for example on page two use APA with "Cuevas et al. 2021" and it should be as you are using it throughout the document with brackets (bracketed format). When using the format with brackets, the author and others are not mentioned (et al. is not used).

In section 2, Materials and methods.

The inclusion criteria should be stated. The explanation should be improved, especially for Table 1.

Tables 1, 2 and 3 are outside the document use the appropriate size.

In table 1, there are three suspensive points, it should indicate what it means or remove them.

Figure 2 should be placed with a better description of the graph, the values they represent should be placed in percentages and the type of text or font should be the same as the document. Figure 3 in the same way.

Figure 4, the legends are incomplete, for example it says "Computer" which is not the same as "Computer Science".

Figure 5, the legends are in lower case.

Figure 7 in dispersion zones should be better explained, especially 28 items if in the other levels there are 29, the sum does not coincide or do not add up?

The way of mentioning figures and tables in the paragraphs should be revised. Figure 2 and figure 8 are not mentioned correctly, remember to use (see Fig. # or see Table #).

Courage and good work.

The work should be improved.

Author Response

(The authors gave the same response as above.)

Reviewer 3 Report

Lacks a little focus on one central aspect – even the technology parts are a little underwhelming, for instance the reporting on lines 298 – 308 is unsurprising.

Equally I don’t think the observation and counter-point in 317 – 320 is ‘controversial’ – I think this just shows divergence between two different studies: this is hardly worth highlighting to my mind.

In 3.2.2 I thought I would find the essence of the paper, but instead I think the paper could do with a more concerted focus on one or two specific lines of enquiry, as it broadly covers: teacher preparedness for using technologies, benefits of using technologies, access to technology, stagnation of people exercising during lockdown, different ways to teach the subject during lockdown using technology, students ICT literacy, games-based approaches, lack of emotional support for those students (and more). All of these were dealt with rather lightly rather than with depth and also through a wide range of contexts (including subject disciplines, subject levels, age group and geographical reach) leading it to feel a little superficial and scattered in attention to a central point.

The finding 333 – 336 is sensible and found elsewhere.

The whole paragraph on Bibliometic law was a bit jarring in terms of paper discussion and flow, for me. ‘Price’s Law’ is suddenly mentioned in 337 and 338 and readers would benefit from an explanation of its relevance and use, while Lotka’s Law is slightly better explained, but Bradford’s Law not at all. Do you assume readers should know these implicitly or should they go and read about them elsewhere to follow the points made? I think these should be brought to the foreground and explained in a distinct paragraph, showing their relationality to the methodology. 

Line 27 delete ‘the’ [different educational systems]

Line 30-31 should read ‘includes problems such as climate change and limited natural resources, reducING inequalities and achieving social inclusion…’

Line 33 delete ‘as’ and ‘that’

Line 99 – would exergame be best defined or given example?

Author Response

(The authors gave the same response as above.)

Reviewer 4 Report

Dear authors,

Thank you for your paper. In reviewing your paper many strong statements have been made from Lines 25 - 45 without any supporting references. In this section, inferences have been made that are not correct regarding influences on educational policies. Educational policies are strongly influenced by the elected Governments, not necessarily a UN statement.

In reading the paper there is an inconsistent use of terms such as "actor" and then "teacher" which poses further questions for me. 

The methodology is unclear and I am surprised that a number of countries that experienced major lockdowns such as Australia and New Zealand had no mention in relation to educational policies and use of technology. 

Many of the paragraphs include statements without any further elaboration or demonstration of what is implied. Major revisions are required throughout the whole paper. 

English is average. 

Author Response

(The authors gave the same response as above.)

Round 2

Reviewer 2 Report

Good job for the modifications in the wording for a better understanding when reading.

The PRISMA flow defines the number of jobs to be included in the study. From this beginning the results are made, page 3, it indicates 86 jobs, it is 100%. Turning to page 4, figure 2 indicates 53.5% + 37.2% + 4.7% = 95.4%, should it be 100%, or is it the cumulative percentage, check this paper:

{Desai, N., Veras, L., & Gosain, A. (2018). Using Bradford's law of scattering to identify the core journals of pediatric surgery. Journal of Surgical Research, 229, 90-95.} 

Should improve the explanation, may be in these places:

1. From the preceding paragraph.

2. The same graph

3. Or the foot of the graph describing in a more extended way what you want to present.

Similarly the other statistical graphs, remember that visually there should be consistency, for example the pie charts should present a totality, which is 100%, but your graphs, figure 4, presents 116.4%. On the other hand, in figure 5, the sum of the graph is 116.3%, it should be 100%, i.e. the total. 

It would not have more modifications in the rest of the document, that is the concern of the revision. The work is of importance in the area and the respective support should be given.

Author Response

We appreciate the suggestions made by the reviewer because they allow us to improve our work. We have made the suggested changes in the text, consisting of the explanation of the data.

The data referring to the current year (2023) are not represented in Figure 2, since the definitive data of publications in this year are not known, and its graphic representation could be misleading (representing at the time of closing the search 4.6% of the total). This information is included both in text and in the title of the figure.

The data have been reviewed and the errors in the Figures 4 & 5 have been corrected, adjusting them to the data that did appear correctly in the text.

Reviewer 4 Report

Dear authors,

Thank you for making all the necessary changes. Well done!

Author Response

We would like to thank the reviewer both for the recommendations he made in the first review for the improvement of the article and for his assessment of the article.

Round 3

Reviewer 2 Report

I agree with the changes.